# LEARNING FROM LABEL PROPORTIONS WITH CONSISTENCY REGULARIZATION

## ABSTRACT

The problem of learning from label proportions (LLP) involves training classifiers with weak labels on bags of instances, rather than strong labels on individual instances. The weak labels only contain the label proportion of each bag. The LLP problem is important for many practical applications that only allow label proportions to be collected because of data privacy or annotation cost, and has recently received lots of research attention. Most existing works focus on extending supervised learning models to solve the LLP problem, but the weak learning nature makes it hard to further improve LLP performance with a supervised angle. In this paper, we take a different angle from semi-supervised learning. In particular, we propose a novel model inspired by consistency regularization, a popular concept in semi-supervised learning that encourages the model to produce a decision boundary that better describes the data manifold. With the introduction of consistency regularization, we further extend our study to non-uniform bag-generation and validation-based parameter-selection procedures that better match practical needs. Experiments not only justify that LLP with consistency regularization achieves superior performance, but also demonstrate the practical usability of the proposed procedures.

## 1 INTRODUCTION

In traditional supervised learning, a classifier is trained on a dataset where each instance is associated with a class label. However, label annotation can be expensive or difficult to obtain for some applications. Take the embryo selection as an example (Hernández-González et al., 2018). To increase the pregnancy rate, clinicians would transfer multiple embryos to a mother at the same time. However, clinicians are unable to know the outcome of a particular embryo due to limitations of current medical techniques. The only thing we know is the proportion of embryos that implant successfully. To increase the success rate of embryo implantation, clinicians aim to select high-quality embryos through the aggregated results. In this case, only label proportions about groups of instances are provided to train the classifier, a problem setting known as learning from label proportions (LLP).

In LLP, each group of instances is called a *bag*, which is associated with a *proportion label* of different classes. A classifier is then trained on several bags and their associated proportion labels in order to predict the class of each unseen instance. Recently, LLP has attracted much attention among researchers because its problem setting occurs in many real-life scenarios. For example, the census data and medical databases are all provided in the form of label proportion data due to privacy issues (Patrini et al., 2014; Hernández-González et al., 2018). Other LLP applications include fraud detection (Rueping, 2010), object recognition (Kuck & de Freitas, 2012), video event detection (Lai et al., 2014), and ice-water classification (Li & Taylor, 2015).

The challenge in LLP is to train models without direct instance-level label supervision. To overcome this issue, prior work seeks to estimate either the individual label (Yu et al., 2013; Dulac-Arnold et al., 2019) or the mean of each class by the label proportions (Quadrianto et al., 2009; Patrini et al., 2014). However, the methodology behind developing these models do not portray LLP situations that occur in real life. First, these models can be improved by considering methods that can better leverage unlabeled data. Second, these models assume that bags of data are randomly generated, which is not the case for many applications. For example, the data of population census are collected on region, age, or occupation with varying group sizes. Third, training these models requires a

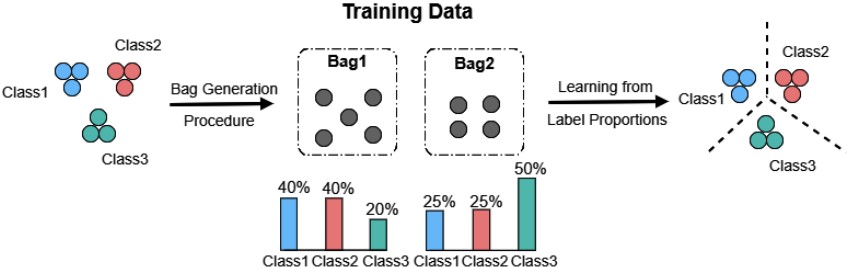

Figure 1: An illustration of multi-class learning from label proportions. Before training, the data are grouped according to a bag generation procedure. During the training stage, we are given bags of unlabeled data and their corresponding proportion labels. The goal of LLP is to learn an individual-level classifier.

validation set with labeled data. It would be more practical if the process of model selection relies only on the label proportions.

This paper aims to resolve the previous problems. Our main contributions are listed as follows:

- We first apply a semi-supervised learning technique, consistency regularization, to the multi-class LLP problem. Consistency regularization considers an auxiliary loss term to enforce network predictions to be consistent when its input is perturbed. By exploiting the unlabeled instances, our method captures the latent structure of data and obtains the SOTA performance on three benchmark datasets.

- We develop a new bag generation algorithm – the K-means bag generation, where training data are grouped by attribute similarity. Using this setup can help train models that are more applicable to actual LLP scenarios.

- We show that it is possible to select models with a validation set consisting of only bags and associated label proportions. The experiments demonstrate correlation between bag-level validation error and instance-level test error. This potentially reduces the need of a validation set with instance-level labels.

## 2 PRELIMINARY

### 2.1 LEARNING FROM LABEL PROPORTIONS

We consider the multi-class classification problem in the LLP setting in this paper. Let $\boldsymbol{x}_i \in \mathbb{R}^D$ be a feature vector of $i$-th example and $y_i \in \{1, \ldots, L\}$ be a class label of $i$-th example, where $L$ is the number of different classes. We define $\boldsymbol{e}^{(j)}$ to be a standard basis vector $[0, \ldots, 1, \ldots, 0]$ with 1 at $j$-th position and $\Delta_L = \{\boldsymbol{p} \in \mathbb{R}_+^L : \sum_i^L \boldsymbol{p}_i = 1\}$ to be a probability simplex. In the setting of LLP, each individual label $y_i$ is hidden from the training data. On the other hand, the training data are aggregated by a bag generation procedure. We are given $M$ bags $B_1, \ldots, B_M$, where each bag $B_m$ contains a set $\mathbb{X}_m$ of instances and a proportion label $\boldsymbol{p}_m$, defined by

$$\boldsymbol{p}_m = \frac{1}{|\mathbb{X}_m|} \sum_{i:\boldsymbol{x}_i \in \mathbb{X}_m} \boldsymbol{e}^{(y_i)}, \quad \bigcup_{m=1}^{M} \mathbb{X}_m = \{\boldsymbol{x}_1, \ldots, \boldsymbol{x}_N\}.$$

We do not require each subset to be disjoint. Also, each bag may have different size. The task of LLP is to learn an individual-level classifier $f_\theta : \mathbb{R}^D \to \Delta_L$ to predict the correct label $y = \arg\max_i f_\theta(\boldsymbol{x})_i$ for a new instance $\boldsymbol{x}$. Figure 1 illustrates the setting of learning from label proportions in the multi-class classification (Dulac-Arnold et al., 2019).

### 2.2 PROPORTION LOSS

The feasibility of the binary LLP setting has been theoretically justified by Yu et al. (2014). Specifically, Yu et al. (2014) propose the framework of *Empirical Proportion Risk Minimization* (EPRM),

proving that the LLP problem is PAC-learnable under the assumption that bags are i.i.d sampled from an unknown probability distribution. The EPRM framework provides a generalization bound on the expected proportion error and guarantees to learn a probably approximately correct proportion predictor when the number of bags is large enough. Furthermore, the authors prove that the instance label error can be bounded by the bag proportion error. That is, a decent bag proportion predictor guarantees a decent instance label predictor.

Based on the profound theoretical analysis, a vast number of LLP approaches learn an instance-level classifier by directly minimizing the proportion loss without acquiring the individual labels. To be more precise, given a bag $B = (\mathbb{X}, \boldsymbol{p})$, an instance-level classifier $f_\theta$ and a divergence function $d_{\text{prop}} : \mathbb{R}^L \times \mathbb{R}^L \to \mathbb{R}$, the proportion loss penalizes the difference between the real proportion label $\boldsymbol{p}_m$ and the estimated proportion label $\hat{\boldsymbol{p}} = \dfrac{1}{|\mathbb{X}|} \sum_{\boldsymbol{x} \in \mathbb{X}} f_\theta(\boldsymbol{x})$, which is an average of the instance predictions within a bag. Thus, the proportion loss $\mathcal{L}_{\text{prop}}$ can be defined as follows:

$$\mathcal{L}_{\text{prop}}(\theta) = d_{\text{prop}}(\boldsymbol{p}, \hat{\boldsymbol{p}}).$$

The commonly used divergence functions are $L^1$ and $L^2$ function in prior work (Musicant et al., 2007; Yu et al., 2013). Ardehaly & Culotta (2017) and Dulac-Arnold et al. (2019), on the other hand, consider the cross-entropy function for the multi-class LLP problem.

## 2.3 CONSISTENCY REGULARIZATION

Since collecting labeled data is expensive and time-consuming, the semi-supervised learning approaches aim to leverage a large amount of unlabeled data to mitigate the need for labeled data. There are many semi-supervised learning methods, such as pseudo-labeling (Lee, 2013), generative approaches (Kingma et al., 2014), and consistency-based methods (Laine & Aila, 2016; Miyato et al., 2018; Tarvainen & Valpola, 2017). Consistency-based approaches encourage the network to produce consistent output probabilities between unlabeled data and the perturbed examples. These methods rely on the smoothness assumption (Chapelle et al., 2009): if two data points $x_i$ and $x_j$ are close, then so should be the corresponding output distributions $y_i$ and $y_j$. Then, the consistency-based approaches can enforce the decision boundary to traverse through the low-density region. More precisely, given a perturbed input $\hat{\boldsymbol{x}}$ taken from the input $\boldsymbol{x}$, consistency regularization penalizes the distinction of model predictions between $f_\theta(\boldsymbol{x})$ and $f_\theta(\hat{\boldsymbol{x}})$ by a distance function $d_{\text{cons}} : \mathbb{R}^L \times \mathbb{R}^L \to \mathbb{R}$. The consistency loss can be written as follows:

$$\mathcal{L}_{\text{cons}}(\theta) = d_{\text{cons}}(f_\theta(\boldsymbol{x}), f_\theta(\hat{\boldsymbol{x}})).$$

Modern consistency-based methods (Laine & Aila, 2016; Tarvainen & Valpola, 2017; Miyato et al., 2018; Verma et al., 2019; Berthelot et al., 2019) differ in how perturbed examples are generated for the unlabeled data. Laine & Aila (2016) introduce the $\Pi$-Model approach, which uses the additive Gaussian noise for perturbed examples and chooses the $L^2$ error as the distance function. However, a drawback to $\Pi$-Model is that the consistency target $f_\theta(\hat{\boldsymbol{x}})$ obtained from the stochastic network is unstable since the network changes rapidly during training. To address this problem, Temporal Ensembling (Laine & Aila, 2016) takes the exponential moving average of the network predictions as the consistency target. Mean Teacher (Tarvainen & Valpola, 2017), on the other hand, proposes averaging the model parametes instead of network predictions. Overall, the Mean Teacher approach significantly improves the quality of consistency targets and the empirical results on semi-supervised benchmarks.

Instead of applying stochastic perturbations to the inputs, Virtual Adversarial Training or VAT (Miyato et al., 2018) computes the perturbed examples $\hat{\boldsymbol{x}} = \boldsymbol{x} + \boldsymbol{r}_{\text{adv}}$, where

$$\boldsymbol{r}_{\text{adv}} = \underset{\boldsymbol{r}:||\boldsymbol{r}||_2 \leq \epsilon}{\arg\max} \, D_{\text{KL}}(f_\theta(\boldsymbol{x}) \| f_\theta(\boldsymbol{x} + \boldsymbol{r})). \tag{1}$$

That is, the VAT approach attempts to generate a perturbation which most likely causes the model to misclassify the input in an adversarial direction. Finally, the VAT approach adopts Kullback-Leibler (KL) divergence to compute the consistency loss. In comparison to the stochastic perturbation, the VAT approach demonstrates the greater effectiveness in the semi-supervised learning problem.

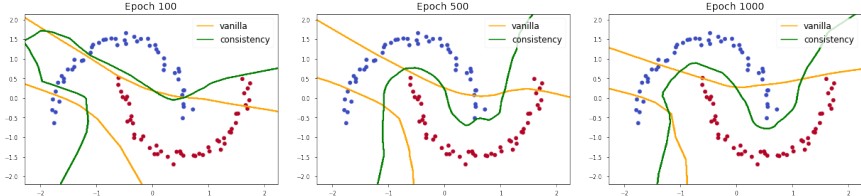

Figure 2: In this toy example, we generate 5 bags, each of which contains 20 data points uniformly sampled from the "two moons" dataset without replacement. The vanilla approach, which simply optimizes the proportion loss, suffers from poor performance as the label information is insufficient. In contrast, the "two moons" can be effectively separated into two clusters by LLP with consistency regularization. Our method enforces the network to produce consistent outputs for perturbed examples, and thus help capture the underlying structure of the data.

## 3 LLP WITH CONSISTENCY REGULARIZATION

With regards to weak supervision, the LLP scenario is similar to the semi-supervised learning problem. In the semi-supervised learning setting, only a small portion of training examples is labeled. On the other hand, in the LLP scenario, we are given the weak supervision of label proportions instead of the strong label on individual instances. Both settings are challenging since most training examples do not have individual labels. To address this challenge, semi-supervised approaches seek to exploit the unlabeled examples to further capture the latent structure of data.

Motivated by these semi-supervised approaches, we combine the idea of leveraging the unlabeled data into the LLP problem. We make the same smoothness assumption and introduce a new concept incorporating consistency regularization with LLP. In particular, we consider the typical cross-entropy function between real label proportions and estimated label proportions. Given a bag $B = (\mathbb{X}, \boldsymbol{p})$, we define the proportion loss $L_{\mathrm{prop}}$ as follows:

$$\mathcal{L}_{\mathrm{prop}}(\theta) = -\sum_{i=1}^{L} \boldsymbol{p}_i \log \frac{1}{|\mathbb{X}|} \sum_{\boldsymbol{x} \in \mathbb{X}} f_\theta(\boldsymbol{x})_i.$$

Interestingly, the proportion loss $\mathcal{L}_{\mathrm{prop}}$ boils down to standard cross-entropy loss for fully-supervised learning when the bag size is one. To learn a decision boundary that better reflects the data manifold, we add an auxiliary consistency loss that leverages the unlabeled data. More formally, we compute the average consistency loss across all instances within the bag. Given a bag $B = (\mathbb{X}, \boldsymbol{p})$, the consistency loss $\mathcal{L}_{\mathrm{cons}}$ can be written as follows:

$$\mathcal{L}_{\mathrm{cons}}(\theta) = \frac{1}{|\mathbb{X}|} \sum_{\boldsymbol{x} \in \mathbb{X}} d_{\mathrm{cons}}(f_\theta(\boldsymbol{x}), f_\theta(\hat{\boldsymbol{x}})),$$

where $d_{\mathrm{cons}}$ is a distance function, and $\hat{\boldsymbol{x}}$ is a perturbed input of $\boldsymbol{x}$. We can use any consistency-based approach to generate the perturbed examples and compute the consistency loss. Finally, we mix the two loss functions $\mathcal{L}_{\mathrm{prop}}$ and $\mathcal{L}_{\mathrm{cons}}$ with a hyperparameter $\alpha > 0$, yielding the combined loss $\mathcal{L}$ for LLP:

$$\mathcal{L}(\theta) = \mathcal{L}_{\mathrm{prop}}(\theta) + \alpha \mathcal{L}_{\mathrm{cons}}(\theta),$$

where $\alpha$ controls the balance between the bag-level estimation of proportion labels and instance-level consistency regularization.

To understand the intuition behind combining consistency regularization into LLP, we follow the $\Pi$-Model approach (Laine & Aila, 2016) to adopt the stochastic Gaussian noise as the perturbation and to use $L^2$ as the distance function $d_{\mathrm{cons}}$ in a toy example. Figure 2 illustrates how our method is able to produce a decision boundary that passes through the low-density region and captures the data manifold. On the other hand, the vanilla approach, which simply optimizes the proportion loss, gets easily stuck at a poor solution due to the lack of label information. This toy example shows the advantage of applying consistency regularization into LLP.

According to Miyato et al. (2018), VAT is more effective and stable than $\Pi$-Model due to the way it generates the perturbed examples. For each data example, the $\Pi$-Model approach stochastically

---

**Algorithm 1** LLP-VAT algorithm

---

**Require:** $\mathcal{D} = \{(\mathbb{X}_m, \boldsymbol{p}_m)\}_{m=1}^M$: collection of bags
**Require:** $f_\theta(\boldsymbol{x})$: instance-level classifier with trainable parameters $\theta$
**Require:** $g(\boldsymbol{x}; \theta) = \boldsymbol{x} + \boldsymbol{r}_{\mathrm{adv}}$: VAT augmentation function according to Equation 1
**Require:** $w(t)$: ramp-up function for increasing the weight of consistency regularization
**Require:** $T$: total number of iterations
    **for** $t = 1, \ldots, T$ **do**
        **for** each bag $(\mathbb{X}, \boldsymbol{p}) \in \mathcal{D}$ **do**
            $\hat{\boldsymbol{p}} \leftarrow \frac{1}{|\mathbb{X}|} \sum_{\boldsymbol{x} \in \mathbb{X}} f_\theta(\boldsymbol{x})$            ▷ Estimated proportion label
            $\mathcal{L}_{\mathrm{prop}} = -\sum_{j=1}^L \boldsymbol{p}_i \log \hat{\boldsymbol{p}}_i$         ▷ Proportion loss
            $\mathcal{L}_{\mathrm{cons}} = \frac{1}{|\mathbb{X}|} \sum_{\boldsymbol{x} \in \mathbb{X}} D_{\mathrm{KL}}(f_\theta(\boldsymbol{x}) \| f_\theta(g(\boldsymbol{x}; \theta)))$   ▷ Consistency loss
            $\mathcal{L} = \mathcal{L}_{\mathrm{prop}} + w(t) \cdot \mathcal{L}_{\mathrm{cons}}$         ▷ Total loss
            update $\theta$ by gradient $\nabla_\theta \mathcal{L}$         ▷ e.g. SGD, Adam
        **end for**
    **end for**
    **return** $\theta$

---

perturbs inputs and trains the model to assign the same class distributions to all neighbors. In contrast, the VAT approach focuses on neighbors that are *sensitive* to the model. That is, VAT aims to generate a perturbed input whose prediction is the most different from the model prediction of its original input. The learning of VAT approach tends to be more effective in improving model generalization. Therefore, we adopt the VAT approach to compute the consistency loss for each instance in the bag. Additionally, to prevent the model from getting stuck at a local optimum in the early stage, we use the exponential ramp-up scheduling function (Laine & Aila, 2016) to increase the consistency weight gradually to the maximum value $\alpha$. The full algorithm of LLP with VAT (LLP-VAT) is described in Algorithm 1.

## 4 EXPERIMENTS

We evaluate our LLP-VAT on three benchmarks, including SVHN, CIFAR10, and CIFAR100. For model selection, we choose hyperparameters using a validation set without individual labels. Lastly, we report the test instance accuracy averaged over the last 10 epochs. The full experiment details are provided in the supplementary material.

### 4.1 UNIFORM BAG GENERATION

For convenience, most LLP works validate their proposed methods with the uniform bag generation where the training data are randomly partitioned into bags of the same size. We evaluate our method using this bag generation procedure with the bag size $n \in \{16, 32, 64, 128, 256\}$. We drop the last incomplete bag if the number of training data is indivisible by the bag size. Table 1 shows the experimental results for the LLP scenario with a uniform bag generation.

In comparison to the vanilla approach, our LLP-VAT significantly improves the performance on CIFAR10 and CIFAR100. This indicates that applying consistency regularization into LLP does help learn a better classifier. As for SVHN, since the test accuracy is close to the fully-supervised performance when the bag size is small, there is no clear difference among three methods. In addition, the results also show that the performance of ROT is unstable and lead us to conclude that the unhelpful pseudo-labels would easily result in a worse classifier. Conversely, our LLP-VAT is more stable and obtains better test accuracy in most cases.

### 4.2 K-MEANS BAG GENERATION

In this section, we further investigate our LLP-VAT in a more practical scenario. We observe that the uniform bag generation barely fits the real-world LLP situation because of following two reasons. First, the real-life data are usually grouped by attribute similarity instead of uniformly sampled. Second, each bag may have different bag sizes, i.e., the distribution of bag sizes is diverse. Consider the

Table 1: Test accuracy with the uniform bag generation. The performance of the vanilla approach with a bag size of one corresponds to the fully-supervised setting.

| Dataset | Method | Bag Size | | | | | |
|---|---|---|---|---|---|---|---|
| | | 1 | 16 | 32 | 64 | 128 | 256 |
| SVHN | vanilla | 95.52 | 95.28 | 95.20 | 94.41 | 88.93 | 12.64 |
| | ROT | - | 95.35 | 94.84 | 93.74 | **92.29** | **13.14** |
| | LLP-VAT | - | **95.66** | **95.73** | **94.60** | 91.24 | 11.18 |
| CIFAR10 | vanilla | 90.64 | 88.77 | 85.02 | 70.68 | 47.48 | 38.69 |
| | ROT | - | 86.97 | 77.01 | 62.93 | 48.95 | 40.16 |
| | LLP-VAT | - | **89.30** | **85.41** | **72.49** | **50.78** | **41.62** |
| CIFAR100 | vanilla | 59.79 | 58.58 | 48.09 | 20.66 | 5.82 | 2.82 |
| | ROT | - | 54.16 | 47.75 | **29.38** | 7.95 | 2.63 |
| | LLP-VAT | - | **59.47** | **48.98** | 22.84 | **9.40** | **3.29** |

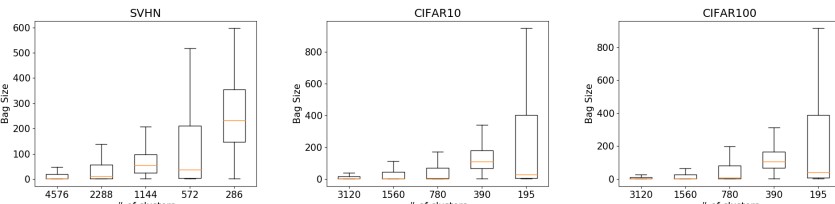

Figure 3: The distribution of bag sizes from the K-means procedure on three benchmarks. When the number of clusters increases, the distribution of bag sizes becomes various.

US presidential election results (Sun et al., 2017), where the statistics of voting results are collected by geological regions (e.g., states). Also, each state have varying number of voters. Therefore, we introduce a new bag generation procedure—the K-means bag generation, where we cluster examples into bags by the K-means algorithm. Although those bags generated from the K-means bag generation are dependent on each other, violating the i.i.d. assumption, this setting is both challenging and worth-studying.

Since we perform experiments on image datasets, it is meaningless to cluster data examples based on RGB pixels. We first adopt the principle component analysis algorithm, which is an unsupervised dimension reduction technique, to project the data into a low-dimensional representation space. This space may capture more important patterns in an images. Then we group the low-dimensional representations of the images following the K-means bag generation procedure. We conduct experiments with the number of clusters $K \in \{3120, 1560, 780, 390, 195\}$ on CIFAR10 and CIFAR100, and $K \in \{4576, 2288, 1144, 572, 286\}$ on SVHN. These numbers are selected to match the number of proportion labels in the uniform bag generation procedure. The distribution of bag sizes generated from the K-means procedure are shown in Figure 3.

For experiments, we do not compare our proposed method to the ROT loss, which needs to estimate individual labels iteratively for each bag. The procedure of the ROT algorithm is time-consuming and cannot be accelerated if bags are of varying sizes. Besides, for the K-means bag generation, there may be some large bags when the value of $K$ is small. Because of the limited computational resource, we take a subsample in each bag if the bag size is larger than the threshold of 256. Particularly, when a large bag is sampled, we randomly sample 256 instances and assign the original label proportions to the reduced bag.

The experimental results of the K-means bag generation are shown in Table 2 and Table 3. Although this scenario violates the i.i.d. assumption, the results demonstrate that it is feasible to learn an instance-level classifier by simply minimizing the proportion loss. Also, our LLP-VAT significantly brings benefits for the k-means bag generation scenario on SVHN and CIFAR10, while showing comparable performance on CIFAR100. Interestingly, the performance of a model is not well-

Table 2: Test accuracy with the K-means bag generation on SVHN.

| Dataset | Method | K | | | | |
|---|---|---|---|---|---|---|
| | | 4576 | 2288 | 1144 | 572 | 286 |
| SVHN | vanilla | 92.07 | 91.16 | 92.00 | 78.70 | **47.16** |
| | LLP-VAT | **93.11** | **91.69** | **93.21** | **82.05** | 46.38 |

Table 3: Test accuracy with the K-means bag generation on CIFAR10 and CIFAR100.

| Dataset | Method | K | | | | |
|---|---|---|---|---|---|---|
| | | 3120 | 1560 | 780 | 390 | 195 |
| CIFAR10 | vanilla | 73.93 | 66.54 | 44.12 | 49.85 | **39.86** |
| | LLP-VAT | **77.43** | **68.01** | **51.04** | **50.22** | 38.27 |
| CIFAR100 | vanilla | **38.65** | **22.16** | **16.07** | **15.47** | 7.82 |
| | LLP-VAT | 37.98 | 21.90 | 15.61 | 15.31 | **8.13** |

correlated with the value of $K$. One possible reason is that we might drop informative bags as we randomly split bags into validation and training.

### 4.3 VALIDATION METRICS

Many modern machine learning models require a wide range of hyperparameter selections about the architecture, optimizer and regularization. However, for the realistic LLP scenario, we have no access to labeled instances during training. It is crucial to choose appropriate hyperparameters based on the bag-level validation error that is computed with only proportion labels. To evaluate the performance at the bag level, we consider four validation metrics: soft $L^1$ error, hard $L^1$ error, soft KL divergence, and hard KL divergence. Their definitions are given as follows. First, we define the output probabilities of an instance as the soft prediction and its one-hot encoding as the hard prediction. For each bag, we then compute the estimated label proportions by averaging these soft or hard predictions. Finally, we use the $L^1$ error or KL divergence to measure the bag-level prediction error.

To investigate the relationship between the instance-level test error and the bag-level validation error, we compute the Pearson correlation coefficient between them on models trained for 400 epochs. The results are shown in Table 4. Surprisingly, we find that the hard $L^1$ error has a strong positive correlation to test error rate on all benchmarks. This implies that it is feasible to select hyperparameters with only label proportions in realistic LLP scenarios. Interestingly, our finding is coherent to Yu et al. (2013). Although their and our works both adopt the hard $L^1$ error for model selection, we focus on the multi-class LLP scenario instead of the binary classification problem they considered. Therefore, we suggest future multi-class LLP works could adopt the hard $L^1$ validation metric for model selection.[1]

## 5 RELATED WORK

Kuck & de Freitas (2012) first introduce the LLP scenario and formulate the probabilistic model with the MCMC algorithm to generate consistent label proportions. Several following works (Chen et al., 2006; Musicant et al., 2007) extend the LLP setting to a variety of standard supervised learning algorithms. Without directly inferring instance labels, Quadrianto et al. (2009) propose a Mean Map algorithm with exponential-family parametric models. The algorithm uses empirical mean operators of each bag to solve a convex optimization problem. However, the success of the Mean Map algorithm is based on a strong assumption that the class-conditional distribution of data is

---

[1]Nevertheless, we do not suggest using our validation metric for early stopping since the correlation is computed after the model converges.

Table 4: The Pearson correlation coefficient between the test error rate and the following validation metrics on benchmarks.

| | Uniform | | | K-means | | |
| --- | --- | --- | --- | --- | --- | --- |
| | SVHN | CIFAR10 | CIFAR100 | SVHN | CIFAR10 | CIFAR100 |
| Hard $L^1$ | **0.97** | 0.81 | **0.81** | **0.99** | **0.75** | **0.67** |
| Soft $L^1$ | 0.83 | 0.33 | -0.50 | 0.90 | 0.61 | 0.26 |
| Hard KL | 0.69 | -0.18 | 0.64 | 0.81 | 0.10 | 0.40 |
| Soft KL | 0.69 | **0.89** | -0.16 | 0.71 | 0.62 | 0.57 |

independent of bags. To loosen the restriction, Patrini et al. (2014) propose a Laplacian Mean Map algorithm imposing an additional Laplacian regularization. Nevertheless, these Mean Map algorithms suffer from a fundamental drawback: they require the classifier to be a linear model.

Several works tackle the LLP problem from Bayesian perspectives. For example, Fan et al. (2014) propose an RBM-based generative model to estimate the group-conditional likelihood of data. Hernández-González et al. (2013), on the other hand, develop a Bayesian classifier with an EM algorithm. Recently, Sun et al. (2017) propose a graphical model using counting potential to predict instance labels for the US presidential election. Furthermore, other works (Chen et al., 2009; Stolpe & Morik, 2011) adopt a k-means approach to cluster training data by label proportions. While some works (Fan et al., 2014; Sun et al., 2017) claim that they are suitable for large-scale settings, both Bayesian methods and clustering-based algorithms are rather inefficient and computationally expensive when applied to large image datasets.

Another line of work adopts a large-margin framework for the problem of LLP. Stolpe & Morik (2011) propose a variant of support vector regression using the inverse calibration method to estimate the class-conditional probability for bags. On the other hand, Yu et al. (2013) propose a procedure that alternates between assigning a label to each instance, also known as *pseudo-labeling* in the literature, and fitting an SVM classifier. Motivated by this idea, a number of works (Wang et al., 2015; Qi et al., 2016; Chen et al., 2017) infer individual labels and updated model parameters alternately. One major drawback of SVM-based approaches is that they are tailored for binary classification; they cannot extend to the multi-class classification setting efficiently.

As deep learning has garnered huge success in a number of areas, such as natural language processing, speech recognition, and computer vision, many works leverage the power of neural networks for the LLP problem. Ardehaly & Culotta (2017) are the first to apply deep models to the multi-class LLP setting. Also, Bortsova et al. (2018) propose a deep LLP method learning the extent of emphysema from the proportions of disease tissues. Concurrent to our work, Dulac-Arnold et al. (2019) also considers the multi-class LLP setting with bag-level cross-entropy loss. They introduce a ROT loss that combines two goals: jointly maximizing the probability of instance predictions and minimizing the bag proportion loss.

## 6 CONCLUSION

In this paper, we first apply a novel semi-supervised learning technique, consistency regularization, to the multi-class LLP problem. Our proposed approach leverages the unlabeled data to learn a decision boundary that better depicts the data manifold. The empirical results validate that our approach obtains better performance than that achieved by existing LLP works. Furthermore, we introduce a non-uniform bag scenario - the K-means bag generation, where training instances are clustered by attribute relationships. This setting simulates more practical LLP situations than the uniform bag generation setting, which is often used in previous works. Lastly, we introduce a bag-level validation metrics, hard $L^1$ error, for model selection with only label proportions. We empirically show that the bag-level hard $L^1$ error has a strong correlation to the test classification error. For real-world applicability, we suggest that multi-class LLP methods relying on hyper-parameter tuning could evaluate their methodology based on the bag-level hard $L^1$ error. One interesting future direction is combining the Mixup. In a nutshell, we hope that future LLP work can further explore the ideas presented in this paper.

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

# A EXPERIMENT DETAILS

## A.1 DATASETS

To evaluate the effectiveness of our proposed method, we conduct experiments on three benchmark datasets, including SVHN (Netzer et al., 2011), CIFAR10, and CIFAR100 (Krizhevsky & Hinton, 2009). The SVHN dataset consists of 32x32 RGB digit images with 73,257 examples for training, 26,032 examples for testing, and 531,131 extra training examples that are not used in our experiments. The CIFAR10 and CIFAR100 datasets both consist of 50,000 training examples and 10,000 test examples. Each example is a 32x32 colored natural image, drawn from 10 classes and 100 classes respectively.

## A.2 EXPERIMENT SETUP

**Implementation details.** For all experiments in this section, we adopt the Wide Residual Network with depth 28 and width 2 (WRN-28-2) following the standard specification in the paper (Zagoruyko & Komodakis, 2016).We use the Adam optimizer (Kingma & Ba, 2014) with a learning rate of 0.0003. Additionally, we train models for a maximum of 400 epochs with a scheduler that scales the learning rate by 0.2 once the model finishes 320 epochs. To simulate the LLP setting, we split the training data by two bag generation algorithms described in Section 4.1 and 4.2. Once completing the bag generation, we then compute the proportion labels by averaging the class labels over each bag. To avoid over-fitting, we follow the common practice of data augmentation (He et al., 2016; Lin et al., 2013) padding an image by 4 pixels on each side, taking a random 32x32 crop and randomly flipping the image horizontally with the probability of 0.5 for all benchmarks.

**Hyperparameters.** We compare our method, LLP-VAT, to ROT (Dulac-Arnold et al., 2019) and the vanilla approach, which simply minimizes the proportion loss. For ROT, we conduct experiments with a hyperparameter of $\alpha \in \{0.1, 0.4, 0.7, 0.9\}$ to compute the ROT loss. Following Oliver et al. (2018), we adopt the VAT approach to generate perturbed examples with a perturbation weight $\epsilon$ of 1 and 6 for SVHN and CIFAR10 (or CIFAR100) respectively. We measure the consistency loss with the KL divergence and a consistency weight of $\alpha \in \{0.5, 0.1, 0.05, 0.01\}$.

**Model selection.** For a fair comparison, we randomly sample 90% of bags for training and reserve the rest for validation. In the LLP setting, since there are no individual labels available in the validation set, we select hyperparameters based on the *hard $L^1$ error* which is computed with only proportion labels. To be more specific, the hard $L^1$ error for a bag $B = (\mathbb{X}, \boldsymbol{p})$ is defined by

$$Err = ||\boldsymbol{p} - \hat{\boldsymbol{p}}||_1, \quad \hat{\boldsymbol{p}} = \frac{1}{|\mathbb{X}|} \sum_{\boldsymbol{x} \in \mathbb{X}} \boldsymbol{e}^{(i^*)},$$

where $i^* = \arg\max_i f_\theta(\boldsymbol{x})_i$ and $\boldsymbol{e}^{(i^*)}$ is the one-hot encoding of the prediction. Lastly, we report the test instance accuracy averaged over the last 10 epochs.

# B CONVERGENCE ANALYSIS OF LLP-VAT

To analyze the convergence performance of LLP-VAT, we plot the instance accuracy on the test set over training epochs. Figure 4 and 5 show the accuracy curve on the test set with the uniform bag generation and the K-means bag generation respectively. As shown in Figure 4 and 5, the experimental results demonstrate the stability of our LLP-VAT. When the training epoch gradually increases, the test instance accuracy goes up quickly and converges in the end.

## B.1 UNIFORM BAG GENERATION

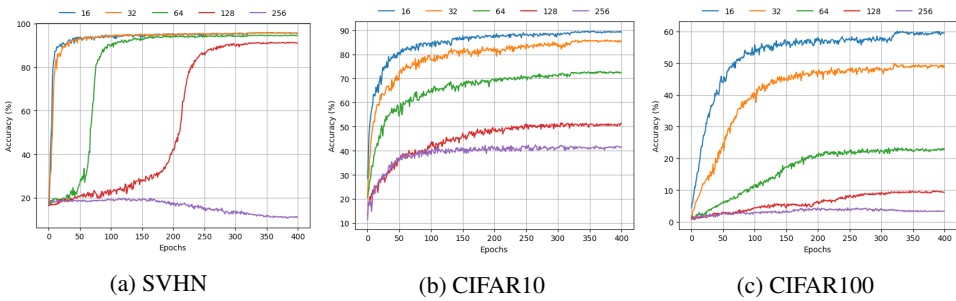

(a) SVHN            (b) CIFAR10            (c) CIFAR100

Figure 4: Evolution of the test accuracy on benchmarks with the uniform bag generation of varying bag sizes.

## B.2 K-MEANS BAG GENERATION

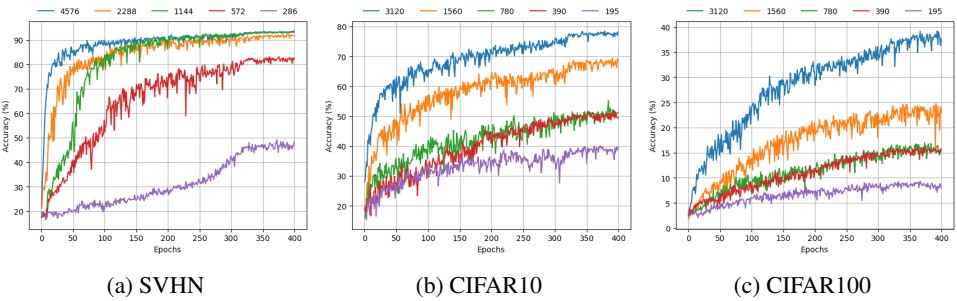

(a) SVHN            (b) CIFAR10            (c) CIFAR100

Figure 5: Evolution of the test accuracy on benchmarks with the K-means bag generation of varying number of clusters.

