# OpenReview forum: "Learning from Label Proportions with Consistency Regularization"
_ICLR.cc/2020/Conference — Reject_

### Official Review · AnonReviewer2 · 2019-10-21
**Official Blind Review #2**

**Rating:** 3

**Review:**

Summary of the paper:  Learning from label proportions (LLP) is an area in machine learning that tries to learn a classifier that predicts labels of instances, with only bag-level aggregated labels given at the training stage.  Instead of proposing a loss specialized for this problem, this paper proposes a regularization term for the LLP problem. The core contribution of this paper is to use the idea of consistency regularization, which has become very popular in semi-supervised learning in the recent years.  The regularization term takes a perturbation of an input sample, and then force the output of the original and perturbed sample to be similar by minimizing  a KL divergence of the two output distributions.  Experiments show the performance of the proposed method under two bag generation settings.  The paper also finds empirically that the hard L_1 has high correlation with the test error rate, which makes it an ideal candidate when the user splits the validation data from training data (meaning there are no ground truth labels for each instances).

Reasons for the decision of the paper:  The proposed consistency regularization term seems to be borrowed directly from semi-supervised learning (SSL) research, and is not really specialized for the LLP problem.  Although the discovery that adding this regularization term for the LLP problem increases generalization performance is novel, using this term for data without ground truth labels in image datasets such as CIFAR10/100 and SVHN itself has low novelty, since these datasets satisfy the smoothness assumption used by consistency regularization methods.  Another issue is that there are only one trial for every experiment, making it hard for the reader to see if the results are statistically significant or not.  For these two reasons, it was hard to give a high score decision for this paper.  However, the paper’s motivation is interesting, and the direction to explore regularization techniques for the label proportion learning problem is important and seems novel for this area.  If the regularization method can be extended in a way that relates to the LLP problem, it would make the paper much stronger.

Other minor comments:

In the experiments, the average test accuracy of the last 10 epochs are reported.  I was curious if the last epochs have better models than earlier epochs.  In many weakly-supervised areas such as noisy labels, it is common that the accuracy goes up very quickly but then gradually decreases.  Does this also happen for the LLP problem?

In J_cons, does the expectation need to be over p?


***After author response:
Thank you for answering my questions!  Although I have a better understanding of the paper now, I still have the same concerns and would like to keep my score.

**Experience Assessment:**

I have read many papers in this area.

**Review Assessment: Checking Correctness Of Derivations And Theory:**

N/A

**Review Assessment: Checking Correctness Of Experiments:**

I assessed the sensibility of the experiments.

**Review Assessment: Thoroughness In Paper Reading:**

I read the paper at least twice and used my best judgement in assessing the paper.

---

> ### Author Response · Authors · 2019-11-14
> **Response to Reviewer #2**
>
> Thank you for your valuable suggestions and detailed insights. We would like to address your concerns as follows:
>
> 1.  We agree about the simplicity of our proposed method, but we view it as an unexplored direction for the LLP problem. In comparison to ROT (Dulac-Arnold et al., 2019), our work focuses on finding a simpler and computationally cheaper regularization technique that can be easily extended to the LLP scenario. Since the ROT approach needs to optimize the model parameters and estimate the pseudo-labels within a bag alternatively, the procedure of ROT is time-consuming. Although we borrow the concept of consistency regularization from semi-supervised learning, our proposed method for LLP is more efficient than ROT. Also, the empirical results demonstrate not only superior performance but also the stability of our proposed method. For example, the ROT method receives 62.93% test accuracy on CIFAR10 with a bag size of 64, whereas the vanilla approach receives 70.68% test accuracy and our approach obtains a better test accuracy of 72.49%. If we look at Table 1, we can observe that the performance of ROT on CIFAR10 and CIFAR100 with medium bag sizes is unstable and lead us to conclude that the unhelpful pseudo-labels would easily result in a worse classifier. Conversely, our LLP-VAT is more stable and obtains better test accuracy in most cases.
>
> 2. The reason why we conduct the experiments on SVHN, CIFAR10, and CIFAR100 is that we follow the standard experiment setting in the previous LLP work (Dulac-Arnold et al., 2019). In terms of large scale and multi-class classification, we think those image datasets are suitable and competitive for multi-class LLP. Besides, to our best knowledge, there are no existing LLP works make the same smoothness assumption for the multi-class LLP problem. Our work first explores a new direction by incorporating consistency regularization into LLP and demonstrate it does help learn a better classifier.
>
> 3. Thank you for your detailed comments and insights. We present the experimental results for one trail on three benchmarks under the computation constraints. However, we repeated several experiments on CIFAR10 and observed that the performance of our method is stable. Currently, we had run more trails on all benchmarks and we would provide more experimental results in the future.
>
> 4. We checked the learning curve of our method, but we did not see the situation where the accuracy goes up quickly at first and decreases gradually. We plot the test curve over training epochs and the results are shown in Figures 4 and 5.  As shown in Figures 4 and 5, the performance of our approach rises steadily and converges in the end. We provide the evolution of test accuracy on all benchmarks with two bag generation procedures in the supplementary material.
>
> 5. Thanks for pointing that, we removed the expected loss J_cons in the paper. Please check the latest revised version. To avoid confusion, we replace all expected loss with regular loss term computed by each bag. Also, we provide a pseudo algorithm of our approach in the revised paper.

---

### Official Review · AnonReviewer3 · 2019-10-24
**Official Blind Review #3**

**Rating:** 6

**Review:**

Summary:
This paper proposes using Consistency Regularization and a new bag generation technique to better learn classification decision boundaries in a Label Proportion setting.  The consistency regularization works to make sure that examples in the local neighbourhood have similar outputs. The authors further use K-means clustering to create a new bagging scenario they use to mimic real-world LLP settings.

Overall, this paper sheds light on how we can use the underlying structure to better be able to make predictions on labels even when we don't have original labels but proportional bags. This is a very interesting problem area and will be interesting to the field.

# For the experiments, a few notes and comments?
1. It's a bit harder to appreciate some of the performance gains here without understanding the error rates around the accuracy. This would be especially good to know for the larger bag sizes where one expects more uncertainty.
2. For each of the datasets, it would have been also enlightening to provide what the normal labelled performance would be as it would give an indication of the limit when the bag sizes get smaller.
3. I am a bit confused by the k-means scenario. With bag sizes similar to the uniform bagging scenario, would the performance not be more as we are already capturing latent structure with the k-means algorithm?
4. Following up on 3, Does this not have implications for actually preserving some privacy?


# Other Comments
5: It would be interesting to see an approach like mixup can be combined with the consistency concept in this case. So what do you expect when you now combine examples and train the underlying mixed bag objective.

**Experience Assessment:**

I have read many papers in this area.

**Review Assessment: Checking Correctness Of Derivations And Theory:**

I assessed the sensibility of the derivations and theory.

**Review Assessment: Checking Correctness Of Experiments:**

I carefully checked the experiments.

**Review Assessment: Thoroughness In Paper Reading:**

I read the paper at least twice and used my best judgement in assessing the paper.

---

> ### Author Response · Authors · 2019-11-14
> **Response to Reviewer #3**
>
> We thank the reviewers for their time and effort in reviewing our paper. We are happy that you find the ideas interesting and grateful for many useful comments.
>
> 1. Thanks for your valuable suggestions. We present the experimental results for one trail on three benchmarks under the computation constraints. However, we repeated several experiments on CIFAR10 and observed that the performance of our method is stable. Currently, we had run more trails on all benchmarks and we would provide more experimental results in the future.
>
> 2. Thanks for your comments. We added the performance of the vanilla approach with the bag size of 1 in Table 1. The test accuracy with a bag size of 1 corresponds to the performance of the fully-supervised setting. As shown in Table 1, we can find that the test accuracy gradually decreases as the bag size increases. This finding is coherent to previous LLP works (Yu et al., 2013; Dulac-Arnold et al., 2019).
>
> 3.4. We provide more information and details for the K-means bag generation in the latest revised version. As shown in Figure 3, we can see that the distribution of bag sizes becomes diverse as the number of clusters K decreases. We find that the model performance with a smaller value of K is superior to or than the model performance with larger bag sizes in some cases. For example, as shown in Figure 3, the median bag size is around 100 when the value of K is 390 on CIFAR100. The condition is similar to that of uniform bag generation with a bag size of 128. However, the test accuracy of the vanilla approach with a bag size of 128 is 5.82%, whereas the model performance with K of 390 gets 15.47%. This result indicates that the performance can be more as we had already captured the underlying structure of data by the K-means bag generation. Nevertheless, we can also find that the performance would decrease and is not well-correlated with the value of K. One possible reason is that we might drop informative bags as we randomly split bags into validation and training. It is challenging and worth-studying to investigate the cross-validation technique for the K-means bag generation. Also, we acknowledge that it is also an interesting direction to study the privacy-preserving issues for LLP with the K-means procedure.
>
> 5. Thanks for your suggestion. Although we didn't apply the Mixup techniques in our work, we agree that it is an interesting future direction to combining this technique into LLP. It is really interesting to think of generating perturbations by mixing the data examples within a bag. Also, we can generate consistency targets by mixing the proportion labels and optimize the proportion loss with mixed bags. We hope that future LLP works can follow our work and further explore new consistency-based methods for multi-class LLP.

---

### Official Review · AnonReviewer1 · 2019-10-28
**Official Blind Review #1**

**Rating:** 3

**Review:**

[Summary]
The paper presents a solution to the problem of learning from label proportion (LLP) by incorporating regularization by data generation motivated by semi-supervised learning.
It is argued that, by consistency assumption, the classification function defined on the data manifold should be a locally consistent mapping (in the sense of bag label proportion) such that the discrepancy between two bags of data points within a small neighborhood should be constraint. This is further combined with the conventional LLP loss (cross entropy between bag label proportion and classification results) to produce the final loss function for learning. For bag data generation, image attributes are leveraged to group similar data points in the feature space. Evaluation is performed on three benchmarks against vanilla solution and ROT loss.

[Comments]
I’m not sure if I fully follow the contribution and several technical details.

 It looks to me that the major contribution claimed is the novel loss function that combines the proportion loss and the consistency loss, but both seem to be from off-the-shelf solutions from literature with slight variation. E.g., J_prop is the standard cross entropy loss (Ardehaly & Culotta (2017) and Dulac-Arnold et al. (2019)), and J_cons is from the vanilla consistency definition. I had a hard time getting the novelty here.

The notion and definition are somehow confusing to me too. What is K in the second line of page 3? Shouldn’t J_prop in page 4 defined as sum of all per-bag losses? In J_cons of pge 4, shouldn’t x under the summation be x_\hat as x_\hat is sampled? The equations should be properly numbered for easy reference.

The use of image attributes in 4.4 for K-means bag generation seems a strong requirement? What kinds of image attributes are used and how are they generated? Looks like this strategy only applies to image classification?

The results reported could be more clear. The gap between the proposed method and vanilla or ROT does not seem quite big in many cases (less than 1% in the best cases in table 1). I’m not sure if these results are convincing or not as statistical significance is unclear.

With all of the above uncertainty, I do not have confidence to have the paper accepted in the current format based on my preliminary assessment.


**Experience Assessment:**

I do not know much about this area.

**Review Assessment: Checking Correctness Of Derivations And Theory:**

I assessed the sensibility of the derivations and theory.

**Review Assessment: Checking Correctness Of Experiments:**

I assessed the sensibility of the experiments.

**Review Assessment: Thoroughness In Paper Reading:**

I made a quick assessment of this paper.

---

> ### Author Response · Authors · 2019-11-14
> **Response to Reviewer #1**
>
> Thanks for your valuable suggestions and detailed comments.
>
> 1. We agree about the simplicity of our proposed method, but we view it as an unexplored direction for the LLP problem. In comparison to ROT (Dulac-Arnold et al., 2019), our work focuses on finding a simpler and computationally cheaper regularization technique that can be easily extended to the LLP scenario. Since the ROT approach needs to optimize the model parameters and estimate the pseudo-labels within a bag alternatively, the procedure of ROT is time-consuming. Although we borrow the concept of consistency regularization from semi-supervised learning, our proposed method for LLP is more efficient than ROT. Also, the empirical results demonstrate not only superior performance but also the stability of our proposed method. For example, the ROT method receives 62.93% test accuracy on CIFAR10 with a bag size of 64, whereas the vanilla approach receives 70.68% test accuracy and our approach obtains a better test accuracy of 72.49%. If we look at Table 1, we can observe that the performance of ROT on CIFAR10 and CIFAR100 with medium bag sizes is unstable and lead us to conclude that the unhelpful pseudo-labels would easily result in a worse classifier. Conversely, our LLP-VAT is more stable and obtains better test accuracy in most cases.
>
> 2. Thanks for pointing that, we corrected the typo mistake of "K" in the paper. We provide the expected loss over bags because we would sample a few bags as a mini-batch and then compute the corresponding mini-batch loss in each epoch. That is, we would not compute the total loss by summation of all bags in the experiments. To avoid any confusion, we replace all expected losses in section 3 with the regular loss terms and provide a pseudo algorithm of our approach in the last revised version.
>
> 3. The K-means bag generation would cluster data examples into bags by the K-means algorithm. This procedure of bag generation doesn't require image datasets, i.e. we can apply it to any dataset. Therefore, the use of image attributes is not the requirement for the K-means bag generation.
>
> 4. Thanks for your valuable suggestion. We present the experimental results for one trail on three benchmarks under the computation constraints. However, we repeated several experiments on CIFAR10 with different random seeds and observed that the performance of our method is stable. Currently, we had run more trails on all benchmarks and we would provide more experimental results in the future.

---

### Author Response · Authors · 2019-11-14
**Paper Updates**

We would like to thank all reviewers for their valuable suggestions and detailed comments. We have updated the paper with the following changes:

- Addressing the typo pointed out by Reviewer #1 in Section 2.
- Adding the pseudo algorithm of our proposed method in Section 3.
- Replacing all expected losses with regular loss terms in Section 3.
- Adding the baseline performance with bag size of one in Table 1.
- Adding Figure 3 for the distribution of bag sizes under the K-means bag generation procedure in Section 4.
- Providing the evolution of the model performance on all benchmarks in Appendix B.

---

### Decision · Program_Chairs · 2019-12-19

**Decision:**

Reject

**Comment:**

After reading the author's rebuttal, the reviewer still hold that the main contribution is just the simple combination of already known losses. And the paper need to pay more attention on the clarity of the paper.